# The overlapping global distribution of dengue, chikungunya, Zika and yellow fever

Ahyoung Lim [1,2], Freya M. Shearer[3,4], Kara Sewalk[5], David M. Pigott [6,7], Joseph Clarke [8], Azhar Ghouse[1,9], Ciara Judge [1,2], Hyolim Kang[1,2,10], Jane P. Messina [11,12], Moritz U. G. Kraemer [13,14], Katy A. M. Gaythorpe [15], William M. de Souza[16], Elaine O. Nsoesie[17], Michael Celone[6], Nuno Faria [18], Sadie J. Ryan [19], Ingrid B. Rabe[20], Diana P. Rojas[20], Simon I. Hay [6,7], John S. Brownstein [21], Nick Golding [3,4,22] & Oliver J. Brady [1,2] ✉

Arboviruses transmitted mainly by *Aedes* (*Stegomyia*) *aegypti* and *Ae. albopictus*, including dengue, chikungunya, and Zika viruses, and yellow fever virus in urban settings, pose an escalating global threat. Existing risk maps, often hampered by surveillance biases, may underestimate or misrepresent the true distribution of these diseases and do not incorporate epidemiological similarities despite shared vector species. We address this by generating new global environmental suitability maps for *Aedes*-borne arboviruses using a multi-disease ecological niche model with a nested surveillance model fit to a dataset of over 21,000 occurrence points. This reveals a convergence in suitability around a common global distribution with recent spread of chikungunya and Zika closely aligning with areas suitable for dengue. We estimate that 5.66 (95% confidence interval 5.64-5.68) billion people live in areas suitable for dengue, chikungunya and Zika and 1.54 (1.53-1.54) billion people for yellow fever. We find large national and subnational differences in surveillance capabilities with higher income more accessible areas more likely to detect, diagnose and report viral diseases, which may have led to overestimation of risk in the United States and Europe. When combined with estimates of uncertainty, these suitability maps can be used by ministries of health to target limited surveillance and intervention resources in new strategies against these emerging threats.

Arboviruses transmitted by *Aedes aegypti* and *Ae. albopictus* mosquitoes in human-amplified cycles (*Aedes*-borne viruses), including dengue, chikungunya, and Zika viruses, and yellow fever in urban settings, represent a rapidly growing threat on a global scale. In recent decades, their contribution to global mortality and morbidity has grown and their geographical range has significantly expanded[1]. Recent epidemiological data from 2023 and early 2024 reveals an alarming surge in dengue cases across endemic countries, such as Brazil, Peru, and Bangladesh, marking the highest incidence years on record[2–5]. Over the

past decade, chikungunya and Zika emerged and spread throughout the Americas, caused increasingly large outbreaks[6,7] and have spread beyond their historic tropical and subtropical limits to areas such as Mediterranean Europe and the southern United States[8,9]. The current burden of yellow fever is likely underestimated[10,11], impeding efficient planning, allocation, and evaluation of vaccination programmes. The recent challenges posed by upsurges in *Aedes*-borne arboviruses, coupled with climate, demographic changes, and increased human global mobility all of which are expected to worsen current trends,

---

underscore the urgent need to enhance our understanding of the current geographical distribution of these diseases and their potential to expand into new regions[12–14]. In 2022, the WHO Global Arbovirus Initiative was launched to develop an integrated approach to risk monitoring and early detection of *Aedes*-borne arboviruses that would leverage their shared mosquito vectors and environmental drivers[15].

Global maps of disease risk are valuable tools for identifying vulnerable populations, guiding surveillance, and maximising the impact and efficiency of surveillance and control efforts. The spatial distribution of a vector-borne disease in humans is influenced by the spatial distribution of humans, vectors, the environmental features that shape transmission efficiency and mobility to spread the pathogen to suitable areas. However direct observation of this spatial distribution is rarely possible due to spatial variation in access to care, accurate diagnosis and transparent reporting. A growing availability of historical disease occurrence data and high-resolution remote sensing data, along with advancements in geostatistical modelling techniques[16], spurred the development of the first generation of global suitability maps for dengue[17], chikungunya[18], Zika[19], and yellow fever[20] as well as their *Aedes* mosquito vectors[21]. However, previous environmental suitability maps have often been challenged by biases in data arising from the spatially varying levels of propensity to detect, diagnose and report infectious diseases across the world, resulting in maps

that represent disparities in surveillance effort rather than, as intended, disease risk.

Furthermore, previous mapping studies have been disease-specific, neglecting the epidemiological similarities between *Aedes*-borne diseases through their shared vector species and environmental drivers. The recent spread of chikungunya and Zika has, so far, mostly been limited to a subset of areas where dengue has already been reported, but it remains unknown if these diseases will follow the same path of global expansion as dengue or whether they will be constrained by disease-specific relationships with climatic and environmental factors[22].

Here we present new global maps for dengue, chikungunya, Zika and yellow fever that incorporate key data updates and methodological innovations. Our approach builds on an existing ecological niche modelling (ENM) framework, and attempts to address the key challenges of observation bias and shared drivers by: (i) explicitly accounting for spatial variation in surveillance capability for detecting and reporting viral infections; and (ii) joint modelling of multiple *Aedes*-borne arboviruses, leveraging similarities of their shared drivers to overcome disease-specific data gaps. We apply this approach to the latest occurrence records (up to March 2024) for dengue, chikungunya, Zika, and yellow fever. As a result, our maps of different *Aedes*-borne arboviral diseases will be directly comparable, providing the

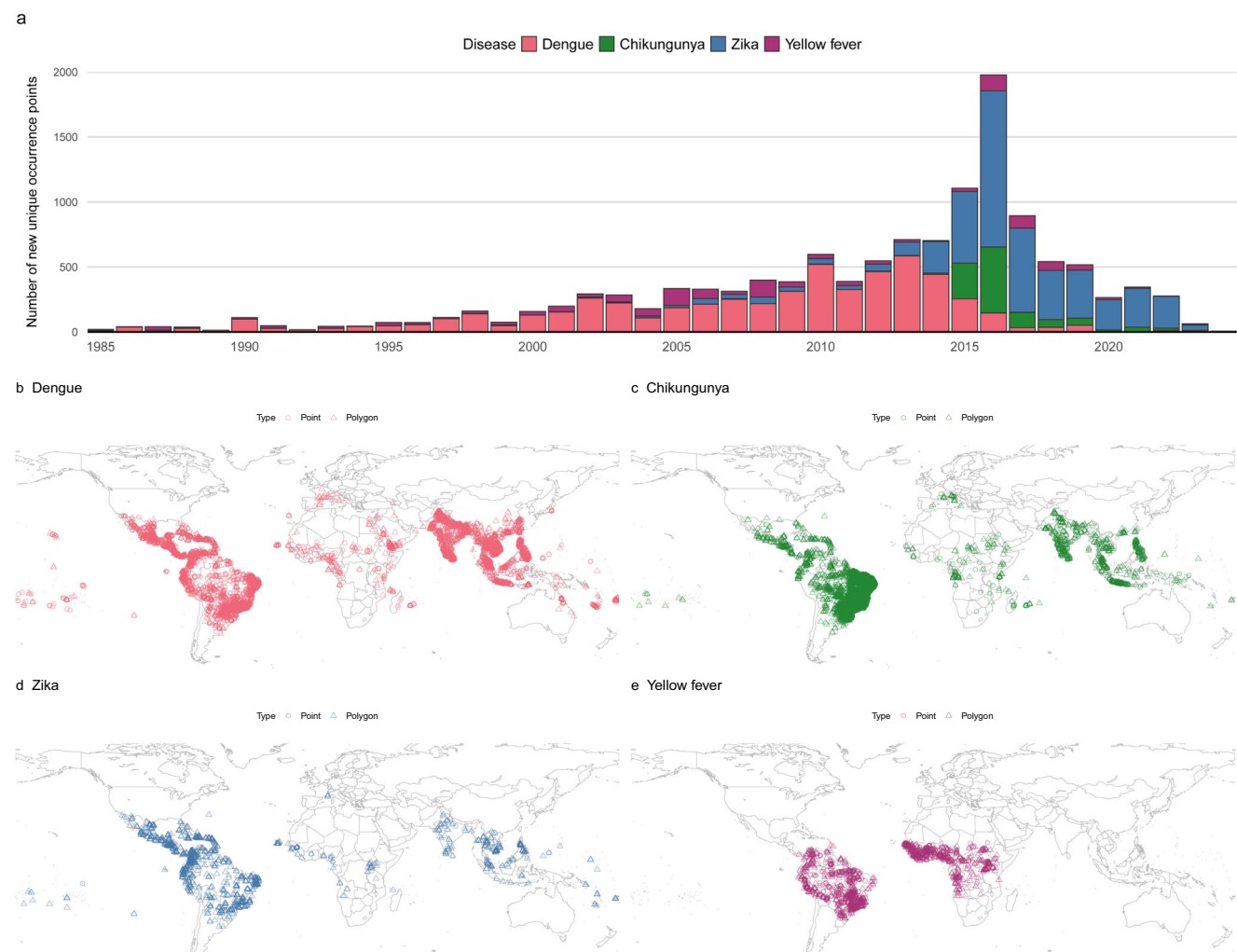

**Fig. 1 | The temporal and spatial distribution of *Aedes*-borne arbovirus occurrence points. a** The global number of new unique occurrence points added each year (i.e. after thinning). Years with sparse (*n* < 100) occurrence records

(1927–1984) are not shown. **b–e** global maps of occurrence data for dengue (**b**), chikungunya (**c**), Zika (**d**) and yellow fever (**e**). The maps were created using public-domain Natural Earth data, accessed through the rnaturalearth package in R[36].

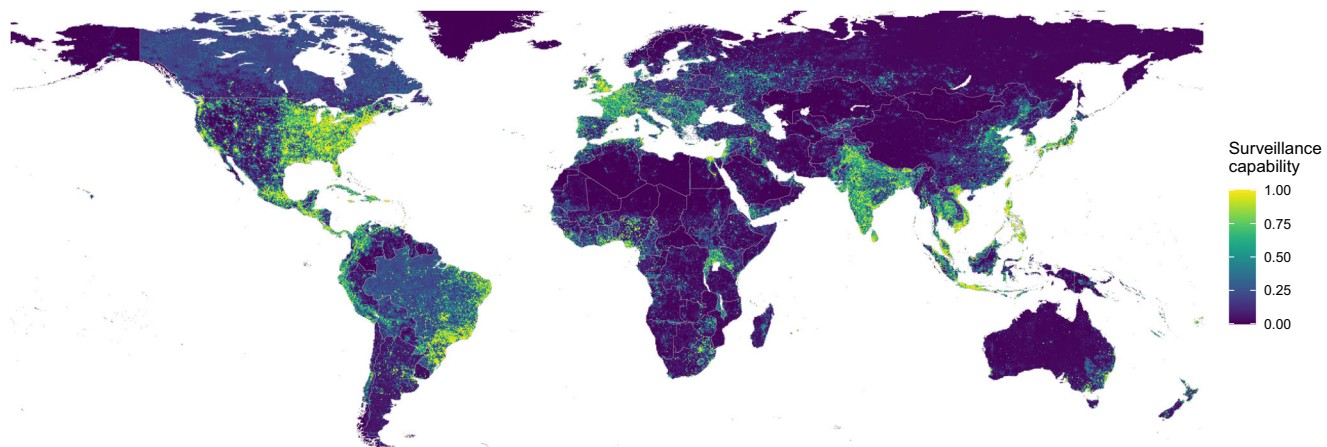

**Fig. 2 | Model-predicted relative surveillance capability for emerging acute viral infectious diseases.** Values close to 1 (yellow) indicate that a viral infection is more likely to be publicly reported. The map was created using public-domain Natural Earth data, accessed through the rnaturalearth package in R[86].

evidence base for a coordinated cross-disease response to their spread and facilitating more effective allocation of resources and interventions across affected regions.

## Results

### The global distribution of occurrence data points

We assembled a total of 58,361 occurrence records between 1927 and March 2024 for dengue, chikungunya, Zika, and yellow fever. These data build on existing occurrence databases[23,24] for each disease adding 23,623 new points for dengue, 12,835 for chikungunya, 4171 for Zika, and 1505 for yellow fever, filling in key gaps like including the spread of Zika after the year 2016 and more recent expansions of chikungunya in South America (Supplementary Fig. 1). The removal of spatial duplicates (thinning) was done by collapsing the presence data over time to prevent repeated occurrences from skewing the geographical representation. This created 13,127 unique locations where dengue (*n* = 5867), chikungunya (*n* = 4727), Zika (*n* = 1138) and yellow fever (*n* = 1395) have occurred across 118, 106, 78 and 34 countries respectively.

The largest increases in the annual number of new unique occurrence points occurred during large outbreaks for chikungunya, Zika and yellow fever (e.g. the 2015–2016 epidemic, Fig. 1a). For dengue, however, the number of novel data points saw a more gradual expansion for decreasing in recent years, suggesting either slowing expansion or stabilisation in the coverage and resolution of dengue reporting. Spatial coverage of the arboviral occurrence data includes all countries that regularly report dengue cases[25], as well as regions where routine *Aedes*-borne arbovirus surveillance is not performed, but *Aedes*-borne arboviruses are known to occur, such as some regions of Africa[26] (Fig. 1b–e).

We found substantial geographic overlap between dengue, chikungunya and Zika occurrence points with 78.3% (3701/4727) of chikungunya and 83.7% (952/1138) of Zika occurrence points falling within 50 km of a dengue occurrence point, suggesting a high degree of overlap in environmental suitability between these different diseases.

The occurrence data for all viral diseases (*n* = 21,700) covers the vast majority of the populated regions of the world. The density of points was highest in high-income countries, including the United States, European nations, and Japan, but also in many countries in the Global South, including Brazil, India, and Thailand (Supplementary Fig. 2).

### Global map of viral surveillance capability

Our predicted map of surveillance capability for acute viral diseases closely mirrors the all viral diseases dataset with higher values in high-income countries but also many Global South countries with known extensive surveillance systems like Brazil, Uganda, Philippines, Vietnam, and Indonesia (Fig. 2). We also find important sub-national variations in surveillance capability, with urban areas generally showing higher levels compared to rural areas, particularly evident in countries like Brazil and India. Examining the surveillance model variable relative importance plots (Supplementary Figs. 3 and 4) shows the importance of shorter travel times to healthcare facilities and a higher GDP for predicting higher reporting probabilities.

The spatial distribution of uncertainty of the predictions from the surveillance capability model shows very low uncertainty (< 0.05) in most areas globally (Supplementary Fig. 5). Validation statistics indicated a high predictive performance of the random forest model evaluated in a 100-fold spatial cross-validation procedure, with an overall AUC of 0.96. Regionally stratified AUC values were good (>0.82) across all regions (Supplementary Fig. 6), with highly populated areas showing higher performance than sparsely populated regions (Supplementary Fig. 7).

### Global maps of arboviral diseases

In agreement with the high spatial overlap in datapoints, our models also suggest that dengue, chikungunya, and Zika share a common global distribution. The arboviral disease variable in the arbovirus model contributes the least to decreased node impurity among all variables tested (Supplementary Fig. 8a). It also has a minimal univariate effect in partial dependency plots (Supplementary Fig. 8b) and does not support disease-specific interactions with other variables as evidenced by near-identical predicted suitability maps (Supplementary Fig. 9). The inclusion of disease-specific thermal traits for Zika also reduced the arbovirus model predictive performance for Zika datapoints (Supplementary Fig. 10), further suggesting commonalities. Due to this, we collapsed the maps for dengue, chikungunya, and Zika into a unified map that draws strengths across datasets from all three diseases (Fig. 3a).

Our model for dengue, chikungunya and Zika predicts a high environmental suitability (hereafter suitability) in many tropical and subtropical regions, with a general higher intensity in South America and Asia but a patchier distribution in Africa (Fig. 3). Our measure of suitability is probability of one or more cases of any of these diseases ever having occurred up to 2024, based on the average environmental conditions of a location over a multi-year period (2010–2020). Areas of predicted high suitability include those where cases and outbreaks have been well documented (e.g. across South America, South East Asia and the Indian sub-continent), but also areas where reporting of arboviral diseases is rare (e.g. West and Central Africa). Such

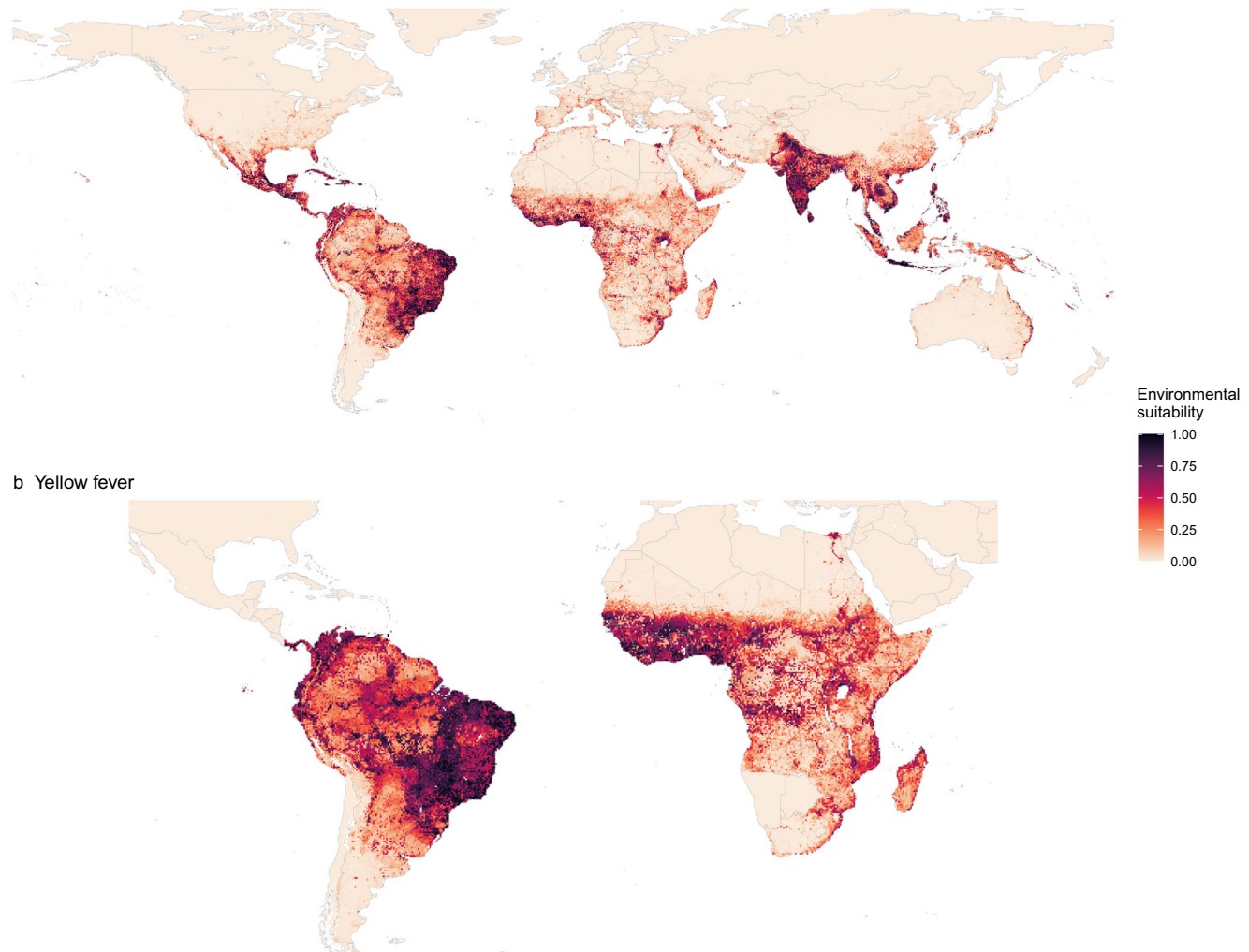

a  Dengue, chikungunya, and Zika

b  Yellow fever

**Fig. 3 | Model-predicted environmental suitability for dengue, chikungunya, and Zika, and yellow fever after accounting for spatial variation in surveillance capacity.** Suitability values represent the probability of one or more cases of diseases having occurred up to 2024, based on the average environmental conditions of a location over the period 2010–2020. Values close to 1 indicate highly suitable conditions for transmission. **a** Areas without a suitable temperature range for transmission have been set to 0 for dengue, chikungunya, and Zika. **b** Areas outside the countries at risk, endemic, or potentially at risk for yellow fever as defined by the WHO yellow fever risk assessment working group[31] have been set to 0. The maps were created using public-domain Natural Earth data, accessed through the rnaturalearth package in R[86].

discrepancies could represent underreporting of arboviral diseases (Fig. 2) or identify areas that could be at risk if the virus were introduced. For Zika and chikungunya, our maps of suitability align closely with areas with reported cases, such as the 2015–2016 Zika emergence in the Americas and longer-term distribution of chikungunya in India, but also include areas where these diseases have not yet been reported including several countries in Asia for Zika and Africa for chikungunya. As expected, we predict risk to be patchier in more temperate areas including the United States, Northeast Asia, Europe, and the Middle East where suitability is concentrated in more foci, usually in urban areas.

The yellow fever map shows high suitability across broad regions of central and eastern Brazil, coastal areas of northern South America and West Africa, with smaller foci of risk across South America and sub-Saharan Africa (Fig. 3b). Here, suitability represents the probability of occurrence of a case of yellow fever in the susceptible population, which in many areas has already been reduced, but not eliminated, due to widespread yellow fever vaccination. Our maps are consistent with

the observed distribution of yellow fever outbreaks since the beginning of the 20th century (Fig. 1e), including more recent expansions in Paraguay and towards the coast in southeast in Brazil[27]. We do, however, also predict suitability in areas where yellow fever cases are rarely (Northeast Brazil, large parts of Southeast Africa) or have never been reported (Argentina and Egypt). These predictions suggest such areas remain at risk despite their distance from areas of active virus circulation.

Uncertainty levels in predictions, as measured by the interquartile range of 100 predictions, are generally very low (<0.05) in most areas for all diseases (Supplementary Fig. 11). We observed generally consistently good predictive performance, with an overall AUC value for each disease being higher than 0.97 (Supplementary Fig. 12). Spatially stratified AUC values were good (>0.79) across all regions, with Africa showing the lowest values for Zika and yellow fever, South America for dengue, and Asia for chikungunya (Supplementary Fig. 13). Urbanisation, population density, and the mean temperature of the coldest month variables consistently improved model accuracy and node

impurity in both arbovirus (dengue, chikungunya, and Zika) and yellow fever models (Supplementary Figs. 8a & 14A). The predicted suitability for *Ae. aegypti* and *Ae. albopictus* (main vectors for dengue, chikungunya, and Zika and yellow fever in urban cycles worldwide) and *Hg. janthinomys* (one of main vectors for enzootic cycle of yellow fever in the Americas) were also important factors in distinguishing areas at risk from those not at risk. The joint model showed comparable performance to individual disease models with slight improvements in some metrics, particularly benefiting Zika, which has sparse data (Supplementary Table 6). Excluding the *Ae. aegypti* covariate from the yellow fever model, on the basis that the vector is not widely involved in transmission in South America[28,29], led to modest decreases in AUC

and minimal changes to the predicted suitability map (Supplementary Fig. 15).

We estimate that globally, 5.66 (95% confidence interval 5.64–5.68) billion people (roughly 73% of the global population in 2022) live in areas that are at-risk of dengue, chikungunya, and Zika, with the vast majority in Asia, followed by Africa and the Americas, encompassing 169 countries (Table 1). We predict that there are 1.54 (1.53–1.54) billion people living in areas at risk of yellow fever, distributed across 54 countries in South America and Africa. The estimated number of people at risk for each country is also provided in the Supplementary Data 1.

**Comparison with previous maps**

Our joint models generally predict a more focal distribution of dengue, chikungunya and Zika than previous maps[17–20], with the most significant reduction in risk observed at the fringes of the global distributions of these diseases (Fig. 4). This refinement better aligns with observed distributions, notably in the United States, western Europe and Northeast Asia, where viral surveillance is robust, yet there are few or no autochthonous case reports. While our maps maintain broad consistency with previous maps, particularly in identifying widespread risks in tropical and subtropical regions (orange in Fig. 4), our analysis also uncovers notable expansions beyond the historical geographical range of these diseases (green in Fig. 4). These areas were not previously identified as at risk but became apparent with the inclusion of additional data from the second half of the 2015–2016 Zika epidemic and subsequent detections in Europe, India, and South East Asia. The expansions into higher latitudes are also notable, particularly in Mexico, Europe, and the Middle East, suggesting including data from more recent expansions is important for observing where expanding distributions of *Aedes* mosquitoes and changing climate may already be driving expansion of arboviral diseases.

**Table 1 | Estimated global population and number of countries at risk for dengue, chikungunya, Zika, and yellow fever**

| Region | Dengue, chikungunya and Zika | Yellow fever |
|---|---|---|
| **Population at risk (in billions)*** | | |
| Global | 5.66 (5.64–5.68) | 1.54 (1.53–1.54) |
| Africa | 1.24 (1.24–1.24) | 1.14 (1.14–1.15) |
| Asia | 3.46 (3.45–3.47) | - |
| Americas | 0.73 (0.73–0.73) | 0.38 (0.38–0.38) |
| Europe | 0.17 (0.17–0.17) | - |
| Oceania | 0.03 (0.03–0.03) | - |
| **Number of countries (sovereign states) at risk**** | | |
| Global | 169 | 54 |

\* Values show mean population estimates with 95% confidence intervals in parentheses. Surveillance capability scores and at-risk population estimates with 95% confidence intervals for each country are available in the Supplementary Data.

\*\* Countries at risk are defined as UN sovereign states where more than 10% of the population is estimated to be living in at-risk areas.

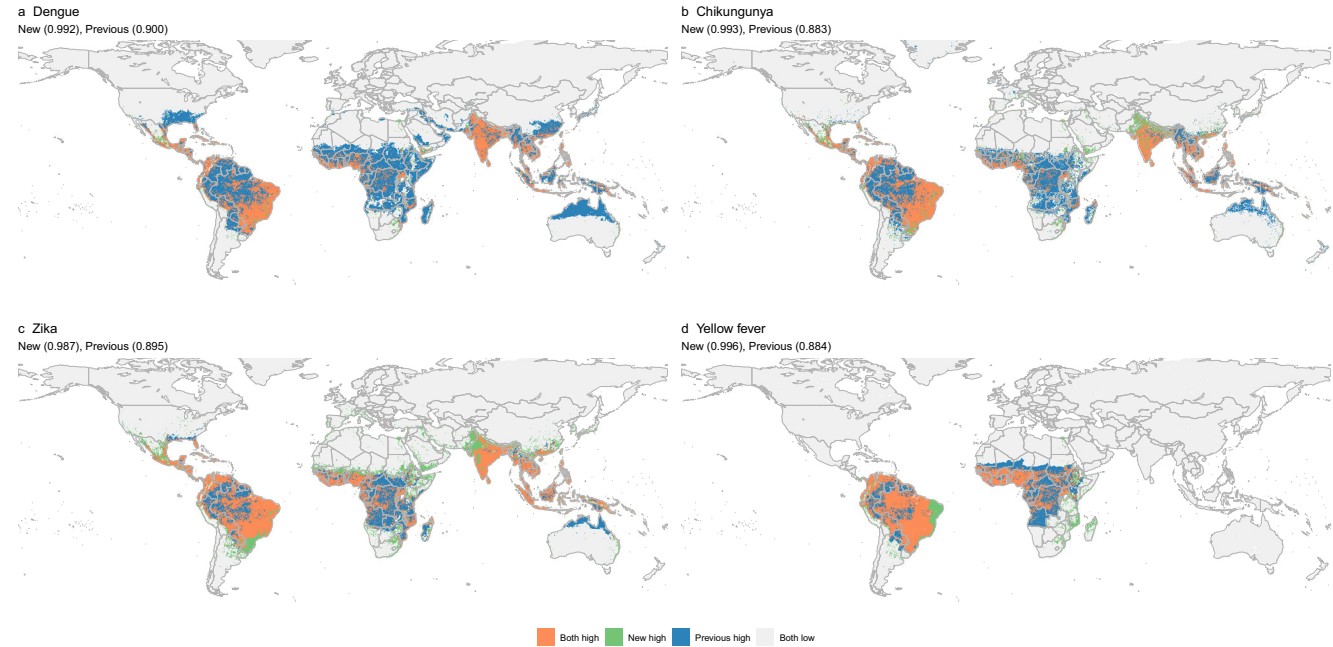

**a  Dengue**
New (0.992), Previous (0.900)

**b  Chikungunya**
New (0.993), Previous (0.883)

**c  Zika**
New (0.987), Previous (0.895)

**d  Yellow fever**
New (0.996), Previous (0.884)

Both high  New high  Previous high  Both low

**Fig. 4 | Comparison of previously published and current suitability maps for arboviral diseases.** Panels show comparisons between earlier published maps and our newly generated maps for (**a**) dengue, (**b**) chikungunya, (**c**) Zika, and (**d**) yellow fever. Previous maps were retrieved from Messina et al.[17] for dengue, Nsoesie et al.[18] for chikungunya, Messina et al.[19] for Zika, and Shearer et al.[20] for yellow fever. Area Under the Curve (AUC) values for our map and previous maps are indicated in parentheses. AUC values measure the model's ability to distinguish between occurrence and background points, with values closer to 1 indicating better predictive performance. The AUC was calculated based on predicted values from both our map and previous maps using presence and background points. Maps were converted into binary format using a threshold that maximised the global sum of sensitivity and specificity. Once binary maps of at-risk areas were generated, we categorised each 5 × 5 km pixel into one of four groups: (1) areas at risk in both our map and the previous maps, (2) areas at risk only in previous maps, (3) areas at risk only in our map, and (4) areas not at risk in either map. The maps were created using public-domain Natural Earth data, accessed through the rnaturalearth package in R[86].

Our map of yellow fever shows a similar distribution in South America, but a patchier distribution in Africa with predicted risk extending to the Southeast of the continent. In previous maps, predictions for coastal Brazil were masked out based on the 2010 expert opinion map by the WHO working group on geographic risk for yellow fever[30]. Following the detection of yellow fever across multiple coastal areas in 2017 and 2018[27] and in consultation with their contemporary equivalent (WHO yellow fever risk assessment working group[31]) their recommendation was now not to mask these areas, but acknowledge that our predictions of risk differ in these areas where they represent the risk of transmission if yellow fever virus were to be introduced.

## Discussion

Global maps of diseases should provide a comprehensive and standardised assessment of risk that can be used to guide health policy and optimise responses to mitigate their impact. One of the priority actions of the Global Arbovirus Initiative at its launch was to develop a global framework to monitor the combined risk of *Aedes*-borne diseases as they share most of the drivers of transmission and an integrated approach is the most effective and logical approach to target integrated arbovirus preparedness, prevention and control activities. Historically, *Aedes*-borne arboviruses have often been mapped separately using different methodologies and approaches[17–20]. This siloed approach has resulted in maps that represent differences in surveillance, testing, and diagnosis as much as the actual distributions of the diseases themselves and under-utilise the similarities between different arboviral diseases. Our approach addresses these by accounting for spatial biases in surveillance and joint modelling of arboviral diseases, respectively to provide more accurate estimates of the global suitability.

Our maps show that recent spread of chikungunya and Zika has been wholly within areas already suitable for dengue. This indicates that any area with dengue is at risk of, or may already have experienced, transmission of Zika or chikungunya viruses. The similarity between the distributions of dengue, chikungunya, and Zika has been obscured in previous maps due to the use of different modelling approaches, which hindered direct comparisons between the diseases. Here, we compared and combined suitability maps of the different *Aedes*-borne diseases by fitting our model to the updated occurrence data for each disease and by using a common joint modelling approach. Given the high potential for misdiagnosis and under-reporting of Zika and chikungunya[32], this combined map provides the best estimate of their distribution and basis for identifying vulnerable populations where prevention, disease detection, and clinical management should be prioritised.

Our analysis addresses the long-standing issue of spatial surveillance biases in the field of global disease mapping by nesting separate models for surveillance and disease risk. Occurrence data inherently has biases towards areas with more robust surveillance systems, such as the USA and Europe, potentially overestimating disease risk in these areas. Separation of surveillance and arbovirus risk models allows more accurate estimates of the relationships between their respective risk factors and leads to more specific and focal predictions of disease risk, particularly in areas that have been historically oversampled like Europe and the USA. These more focal and refined maps better align with the sporadic nature of dengue, chikungunya and Zika outbreaks in Europe and USA and enable a more targeted approach to detecting and responding to arboviral introduction.

Quantifying spatial uncertainty is increasingly important in risk mapping to show where additional data collection will be most valuable. Our maps of surveillance capability (Fig. 2), model prediction uncertainty (Supplementary Fig. 5), and spatial predictive performance (Supplementary Fig. 7) can be used to fill gaps in surveillance, detect regional outliers where risk factors act differently, and identify further risk factors that may better explain the distribution of these

diseases respectively. This comprehensive assessment of uncertainty serves as a useful starting point for collaboration with researchers and country ministries of health to identify new data sources and targeted data collection activities to prospectively validate the suitability maps. Validation by stakeholders in affected regions will both improve the maps but also build engagement with the maps, increasing the likelihood that they can usefully inform surveillance and control policy changes.

A key limitation of these maps is that they show all areas suitable for transmission not where these viruses are currently circulating which is likely to be more constrained by viral dispersal. Pairing these maps with estimates of human mobility from endemic areas could allow more specific estimates of areas currently experiencing transmission. Targeted testing for disease in areas identified as suitable but with no recent datapoints is also important to differentiate between under detection and true absence. Past and future changes in climate, urbanisation, surveillance capacity and spread of arboviral vectors could all change the resulting risk maps. Due to data sparsity at the global level it is necessary to aggregate data over multiple years, limiting our ability to capture these changes in global risk over time. While previous work has found that such changes will only have minimal and gradual impacts on the environmental niche for dengue, tipping point events, such as invasion of *Ae. aegypti* into mainland Europe could lead to more rapid changes in risk necessitating updating of these maps. Our surveillance model may also be influenced by the spatial distribution of each of the viral diseases used in the training data, limiting our ability to isolate pure surveillance effort. This model also assumed the equivalent detection probabilities across different viral infections despite their different clinical presentations and methods of diagnosis. To minimise these effects we chose a broad range of viral diseases with different global distributions, defining surveillance capability as the wider investment in detecting, diagnosing, and reporting any circulating pathogens, but future studies could incorporate more direct measures of surveillance, like surveillance system assessments[33] or comparisons between reported case data and serologically-detected infections[34]. Our surveillance model did not include data on COVID-19 and the sustaining gains during the pandemic, which has likely since changed the distribution of surveillance and improve the testing and sequencing capacities worldwide, as well as disrupted transmission dynamics[34] and yellow fever vaccination efforts[35]. Our yellow fever model also only included vaccination coverage as of 2020 and did not capture recent vaccination activities[36]. Finally, our models did not include other vector mosquito species that are known to transmit arboviral diseases in enzootic and epizootic cycles in different regions, including *Ae. africanus*, mosquitoes from the genera *Culex*, *Anopheles*, *Haemagogus* and *Sabethes*[37], which may improve map estimates, particularly for yellow fever.

Here, we produce new global high spatial-resolution maps of environmental suitability for dengue, chikungunya, Zika, and yellow fever and provide up-to-date insights into the evolving landscape of these diseases. Our approach fills key data gaps, addresses issues of spatial biases in surveillance and shares information between *Aedes*-borne viral diseases. The overlap in global environmental suitability among these diseases highlights the interconnected nature of arboviral transmission and underscores the need for integrated and coordinated responses. Our suitability maps offer a starting point for national to local-scale risk assessments over the short to medium term, helping to target interventions and surveillance efforts. These maps should be used in combination with local risk mapping studies that utilise fine-scale spatial information on arboviral disease cases and mosquito counts to further target locally. New data generated from such efforts could be used to iteratively update these maps to improve their contemporariness and operational relevance, contributing to more effective disease control efforts worldwide.

## Methods

To map environmental suitability for *Aedes*-borne arboviral disease, we applied an ENM framework which required: (i) occurrence data; (ii) a set of climatic, environmental and socioeconomic covariates that are known to influence the transmission of these viruses or population dynamics of their vectors; and (iii) a statistical model that parameterises a function which describes the predicted probability of disease being present in each spatial location. The resulting model produced 5 × 5 km spatial-resolution global maps of probability of occurrence of dengue, chikungunya, Zika and yellow fever given long-term average environmental conditions.

### Arbovirus occurrence data

The primary epidemiological data included in our analysis was occurrence data, defined as a unique geographic location where one or more cases of a particular disease been reported at any point in time[23]. We extracted existing occurrence data from previous publications on dengue[17], chikungunya[18,38], Zika[19], and yellow fever[20]. Additional searches were conducted using ProMED mail (http://www.promedmail.org) reports between the period 2015–2022 for chikungunya and Zika to fill important temporal gaps following previously established protocol[23,39]. Despite potential variations in coverage and lack of precise geopositioning, ProMED remains valuable for its comprehensive and timely[40,41] tracking of global outbreaks of emerging infectious diseases, supporting traditional surveillance efforts effectively. We also searched through epidemiological bulletins of the ECDC[8,42–44], WHO regional outbreak updates[45,46] and included occurrence data for countries where diseases had not been reported in the past, but had occurred between 2016 and March 2024 (e.g., dengue in Ibiza, Spain and chikungunya in Uruguay), or where occurrence data were available at the sub-national level (e.g., yellow fever in Bolivia, Brazil, and Peru). Finally, occurrence data for dengue, chikungunya, Zika, and yellow fever from HealthMap platform (www.healthmap.org) was extracted and inspected by cross-referencing with the peer-reviewed literature[47–51] and the epidemiological bulletins. Additional data uncovered through cross-referencing with these sources was incorporated only if they related to subnational levels. Detailed information about the data sources and the number of occurrence data extracted from each source are presented in Supplementary Table 1.

We included occurrence data in two forms: point and polygon. Point data represents infection in a specific location with a minimum resolution of 0.05 degrees which represents the highest available resolution of all covariate datasets. Polygon data represents occurrence somewhere within an administrative unit area, but with the exact location unknown. With the exception of small island nations, all polygon data is at the first or second administrative unit. Both point and polygon data underwent a spatial standardisation ("thinning"[52,53]) where repeated reports from the same location, irrespective of their reporting time, were removed. This reduced the influence of arbitrary variations in reporting frequency between locations (e.g. monthly vs annually) that would have increased bias in areas that choose to report more frequently. Like other species and disease mapping studies, our approach assumes "niche conservatism", i.e. the environmental limits that define where transmission is possible, and probable do not change substantially over time. This means that the best estimate of this conserved niche comes from aggregating data over as many years as possible, as opposed to just using contemporary data. The final standardised dataset included 13,127 records, comprising 5337 point locations and 7790 polygon locations. The total number of occurrence data before and after thinning and for each disease are shown in Supplementary Table 2.

### Occurrence data for other viral diseases

To attempt to account for spatial differences in surveillance[17] (see section "Ecological niche model") we assembled a dataset that included occurrence data for other viral diseases from the HealthMap platform (www.healthmap.org). HealthMap is a tool for tracking infectious disease events in 15 functional languages by aggregating data from diverse informal online sources, including news media and social media posts, and employs machine learning algorithms to classify and filter these reports, ensuring relevance and reducing noise[54]. For this analysis, HealthMap data was extracted from a MySQL database using structured query to obtain viral disease occurrences between 2006 and 2019. We carefully reviewed the list of viral infections to identify those under dedicated surveillance programmes and excluded occurrence points for Ebola, HIV/AIDS, measles, and polio, as these may lead to an overestimation of surveillance capability in certain regions, particularly in West African countries. This resulted in a total of 338,005 records, representing occurrence reports for viral diseases that cause acute febrile illness, primarily diagnosed by serology and polymerase chain reaction (PCR). The final standardised dataset after thinning included 21,700 records. The full list of these diseases and their counts before and after thinning is shown in Supplementary Table 3.

### Background points

No globally representative survey data are available for these diseases, making classical geostatistical models and their assumptions of unbiased sampling from a clear denominator population, unsuitable[16]. We instead used a "presence-background" or "presence-only" modelling approach[55] where observed 'presence' points are supplemented with randomly generated background (absence) across all land surfaces. These background points represent areas where we assume the disease is absent, providing a contrast to the observed presence points in the model. To reduce label-imbalance which can lead to bias, the number of background points selected was proportionate to the number of dengue, chikungunya, Zika and yellow fever occurrence points separately in order to maintain a 1:1 ratio between presence and background points with each disease (sometimes referred to as "down-sampling[56]").

### Covariates

We included global raster layers associated with a range of factors hypothesised to be associated either with transmission of *Aedes*-borne arboviruses or with capacity to detect, diagnose and report cases. Our choice of covariates was based on a previous systematic review of arbovirus risk mapping studies[57]. Covariate data sources were prioritised by those that gave the highest spatial resolution and covered the 2010-2020 time period where most of our occurrence point data are concentrated. Further details of coverage and resolution of each covariate are provided in Supplementary Table 4. Where multiple observations were available across multiple years, mean values for each pixels were calculated to produce a synoptic raster layer representing the average over time period covered. Most of the covariates were available at 0.05 degree (~5 × 5 km at the equator) resolution or higher. For covariates available only at a national scale (e.g., treatment seeking, child mortality, government effectiveness and physicians density), we computed the average value for each World Bank income group and assigned these values to countries with missing data, thus ensuring a globally complete raster layer.

Covariates were resampled to a consistent 0.05 degree grid with a common extent and land/sea mask with lakes and major water bodies removed. The log transformation was optionally applied based on the distributions of each covariate, and all covariates were scaled and centred to have a zero mean and variance of 1. Covariate values for each occurrence point were extracted with mean values across administrative unit areas used for polygon data. Maps of each covariate layer are provided in Supplementary Figs. 16 and 17.

## Covariates for surveillance model

To explicitly model spatial variation in the probability of reporting of an arboviral disease (if present in a given location), we considered nine covariates that are known to be related to the likelihood of detection, diagnosis, and reporting of acute viral infections: (i) gross domestic product (GDP) (5 ×5 km resolution and aggregated national level)[58]; (ii) the fraction of urban land[59]; (iii) travel time to healthcare facilities by walk[60]; (iv) travel time to cities (>50,000 people, any travel mode)[60]; (v) treatment-seeking for fever in children under five years old[61]; (vi) child mortality under five years old[62,63]; (vii) physicians density[62,64]; and (viii) government effectiveness[62,65].

## Covariates for arbovirus model

We included nine covariates in our arbovirus model that reflect known drivers of transmission and vector dynamics: (i) temperature suitability for dengue virus transmission[66]; (ii) mean temperature of the coldest month[67] (iii) annual cumulative precipitation[67]; (iv) Normalised Difference Vegetation Index (NDVI)[68]; (v) Dynamic Habitat Indices (DHI)[69]; (vi) predicted suitability for *Ae. albopictus*[69]; (vii) predicted suitability for *Ae. aegypti*[69]; (viii) GDP (aggregated at national level)[58] and (ix) human population density[70]. Three additional covariates were included in yellow fever model: (i) predicted suitability for *Haemagogus janthinomys* in South America[71]; (ii) distribution of non-human primates (NHPs)[20] and (iii) yellow fever vaccination coverage[72]. Predicted suitability for *Ae. albopictus* was not included in the yellow fever model.

## Ecological niche model

**Approach.** We used a machine learning model which has previously proven useful for global environmental suitability mapping applications[57], including dengue[17,39], chikungunya[18], Zika[19], and yellow fever[20], as well as the global distribution map of *Aedes* vectors[21]. Specifically we used a down sampled random forest (RF) approach that balances the presence and background points at a 1:1 ratio, as RF model has shown to outperform many other machine-learning approaches for the modelling of presence-only data across a range of examples[56,73].

Our approach involves the development of two separate models: a surveillance model and an arbovirus model. The surveillance model aims to capture the between and within country differences in the long-term average probability that a person infected with an acute viral infection seeks treatment, is correctly diagnosed and is reported in the public domain. It should be noted that such estimates may not correlate with ability to detect unfamiliar newly emerging diseases but will represent longer-term differences in surveillance capacity once locally circulating diseases have been appropriately characterised. The predictions from this surveillance model are then used as an offset in the arbovirus model, allowing the arbovirus model to attribute spatial variations in arbovirus occurrence data solely to drivers of transmission risk.

The surveillance model uses occurrence data on all emerging acute febrile diseases including arboviral diseases (Supplementary Table 3) with an equal ratio of randomly sampled background points, following a uniform distribution across the global landscape. The model formula is as follows, with covariates listed in the same order as in Supplementary Table 4.

$$occ_{viral_i} \sim \text{Bernoulli}(P(occ_{viral_i}))$$
$$\text{logit}(P(occ_{viral_i})) = f(GDP_i, GDP_{National_i},$$
$$Urban_i, travel_{health_i}, travel_{cities_i},$$
$$treatmentseeking_i, childmortality_i, goveffectiveness_i, physician_i)$$
$$(1)$$

$P(occ_{viral_i})$ is the location-specific probability that a record $i$ is an occurrence record of an emerging acute febrile disease, rather than a

background point, and $f$ denotes the non-linear and interacting function of all covariates at the same spatial location as record $i$.

The arbovirus model includes occurrence data for all arboviral diseases (dengue, chikungunya, Zika and yellow fever) combined and includes arboviral disease as a categorical variable ($arbovirus_i$). This allows the model to share information between arboviral diseases, but also generate distinct relationships between risk of each disease and the environmental covariates if they are justified[74], e.g. different vector competencies for different viruses. The overall arbovirus model equation is as follows:

$$\text{logit}(P(occ_{arbovirus_i})) = f(arbovirus_i,$$
$$Temp_i, Tcold_i, Precip_i, NDVI_i, DHI_i,$$
$$aegypti_i, albo_i, Haemagogus_i, NHP_i, \quad (2)$$
$$GDP_{National_i} Urban_i, Pop_i) +$$
$$\log(P(occ_{viral_i}) * (1 - Vaccine_i))$$

where $\log(P(occ_{viral_i})*(1 - Vaccine_i))$ is an offset term. Two different versions of the arbovirus model are fit using the same data but with different covariates: i) an arbovirus model for dengue, chikungunya and Zika which omits covariates from $NHP_i$ and $Haemagogus_i$ and sets $Vaccine_i$ to 0 and ii) a yellow fever arbovirus model that omits the *Ae. albopictus* covariate($albo_i$).

The use of a modelled offset for spatial sampling bias based on detections of a wide range of other related diseases is analogous to the commonly used Target Group Background approach[75], but further enables us to explicitly map the estimated spatial variation in reporting rates. Unlike other approaches that have been proposed in ecology to infer spatial bias in a single model[76], the two-stage approach we employ separates the modelling of surveillance and disease processes. This distinction minimises the potential for overlap in the effects of covariates (e.g. urbanness and GDP) that influence both surveillance and disease dynamics, which is particularly important for the arboviruses we consider.

**Spatial block cross-validation.** Randomly repeated cross validation may lead to an over-optimistic model performance metrics if the independence assumption between training and test data is violated. This is especially common when dealing with spatially structured (clustered) data where training observations are often in close proximity to test observations, introducing spatial autocorrelation[77]. To maintain independence between training and test data, a spatial block 100-fold cross-validation was employed to assess the overall and spatially stratified predictive performance of the model. This approach considers the spatial dependence structure of the data when estimating predictive performance, resulting in a final metric that is typically lower than would be returned by conventional cross-validation procedures. Initially, a grid of ~1000 equal-sized square blocks, each with a size of 500 x 500 km, was generated using the cv_spatial function from the "blockCV" package[78]. Each block was then randomly assigned to 100 roughly equal-sized folds, maintaining the balanced number of presences versus background records in each fold. In each iteration, we reserved one-fold (1%) for validation, which was kept out of the model fitting process (99%) and only used to validate predictions ('out-of-sample' validation). By cycling through the data, each fold was used as a validation fold exactly once and then we averaged model performance statistics across the 100 repeated runs, using the area under the curve (AUC) of the receiver-operating characteristic plot as a metric of goodness-of-fit. AUC is a confusion matrix-based model performance metrics that measures how well the model can discriminate presence from background points[79].

**Spatially stratified AUCs.** To understand and visualise the model's performance in different parts of the world, the spatial map of AUC

values was produced using a geospatial approach. We calculated AUC values for a set of validation polygons that are defined by a 250 km radius around each presence or background (PB) point. ROC curves and AUC statistics were then calculated within each of these polygons which was summarised using the coordinates of the centroid for visualisation purposes. The resulting values were then averaged to derive a comprehensive AUC value for each specific PB point. Additionally, regionally stratified AUCs are calculated by averaging the AUC values of PB points within regional boundaries. This iterative approach was chosen to balance spatial continuity of the map while mitigating the impact of small spatial units on AUC values.

**Prediction and uncertainty levels.** To increase the robustness of model predictions and quantify model uncertainty, we obtained 100 predictions by using RF models that were iteratively calibrated during spatial cross-validation as described above. Each of the 100 fitted sub-models predicted environmental suitability on a scale from 0 to 1. Predictions were made separately for each disease, by updating the "arboviral disease" variable (see section "Ecological niche model") to represent different diseases. To generate the final prediction raster, a weighted average prediction was calculated for each $5 \times 5$ km pixel based on the following equation:

$$P_k = \frac{\sum_{i=1}^{n}(p_{k,i} \times w_i)}{\sum_{i=1}^{n} w_i} \qquad (3)$$

where $P_k$ represents the weighted average of predicted risk for pixel $k$, $p_{k,i}$ represents the predicted risk for pixel $k$ derived from the $i$-th sub-model, $w_i$ represents the AUC of the $i$-th sub-model, and $n$ is total number of sub-models (here $n = 100$). We calculated the interquartile range (IQR) for 100 model predictions at each location to quantify uncertainty. We also generated 1000 bootstrap samples for weighted average prediction for each pixel and determined the confidence intervals by computing 2.5th and 97.5th percentiles from their distributions.

**Variable importance and partial dependence plots.** The metrics employed for variable importance included mean decrease in accuracy and mean decrease in node impurity (measured by the Gini index). The variable importance values were extracted and normalised by dividing each value by the sum of all feature importance values and averaged across 100 folds. Partial dependence values for each model were also extracted using the "pdp" package and aggregated across 100 models by calculating the mean and the 95% confidence interval.

**Masking**
Following previous global mapping studies[17,21], to limit extrapolation of risk predictions far beyond their original fitting datasets we post-hoc mask out (set risk to 0) model-based risk predictions in areas where alternative forms of evidence suggest that transmission is extremely unlikely. This included where temperature profiles were estimated as too cold to allow mosquitoes to survive long enough to complete the extrinsic incubation period of the dengue virus[66]. Temperature suitability values were calculated using the mean temperature of 2010–2020[67] and converted to binary range maps using threshold of $14/365 = 0.04$ as they were unlikely to support transmission over the two week serial interval of autochthonous arboviral disease transmission. Environmental suitability predictions for dengue, chikungunya and Zika in areas with temperature suitability values of less than 0.04 were set to 0. Risk predictions for yellow fever were set to 0 in areas outside the countries at risk, endemic, or potentially at risk for yellow fever as defined by the WHO yellow fever risk assessment working group which excluded countries outside of South America and Africa[31]. Masking layers and unmasked versions of environmental suitability maps are provided in Supplementary Figs. 18 and 19 to show the regions of the world affected by the masking process.

**Estimating distribution and population at risk**
The continuous suitability maps were converted into binary distribution maps using a threshold value above which an area was classified as at-risk. We defined the threshold as the suitability value that maximised the global sum of sensitivity and specificity when compared the original presence and background points. Separate thresholds were generated for each disease, with the following suitability values: dengue (0.37), chikungunya (0.21), Zika (0.14), and yellow fever (0.32), to produce individual binary maps. An aggregate binary map for dengue, chikungunya and Zika was then created by combining these individual binary maps from the three diseases, such that any area predicted to be at risk for any of these diseases was considered at risk. Using these binary maps and the global population grid[70], we calculated population at a global, continent and country (UN member state; 193 in total) level. For the purpose of estimating total countries at risk, a country was only included if more than 10% of its total population was identified as being at risk of disease.

**Sensitivity analyses**
To test if the maps for Zika were improved with a disease-specific temperature suitability covariate, the dengue temperature suitability covariate was replaced with a Zika temperature suitability covariate based on temperature relationships from Ryan et al.[80] The improvement of the maps was assessed using a 50-fold block cross validation approach comparing overall and regionally-stratified model performance metrics (AUC, sensitivity and specificity) of the base model (as presented in the section "Ecological niche model") against models with alternative specifications for the relationship between temperature and transmission risk[80–83].

Given that *Ae. aegypti* play a minimal role in modern transmission of yellow fever virus in South America, where transmission primarily occurs through sylvatic spillover rather than urban cycle[28,29], an alternative version of the yellow fever arbovirus model was fit without the *Ae. aegypti* covariate. The improvement of this model was assessed by using a 50-fold block cross validation approach and comparing overall and South America-specific model performance metrics with those of the base model.

To assess the benefits of joint modelling over disease-specific models, we re-ran individual models for dengue, chikungunya, Zika, and yellow fever separately and compared the model performance metrics (AUC, sensitivity, and specificity) with those of the joint model. Model performance was evaluated using 10-fold cross-validation with metrics calculated based on a cutoff value where the sum of sensitivity and specificity is maximised.

**Comparison between diseases and with previously published maps**
We obtained previously published suitability maps from Messina et al.[17] for dengue, Nsoesie et al.[18] for chikungunya, Messina et al.[19] for Zika, and Shearer et al.[20] for yellow fever. These maps were chosen because they represent recent global predictions of disease risk and also utilise occurrence data and species distribution modelling approaches[57], making a more useful assessment of the advances made in this work. The same threshold calculation above was performed for the previous suitability maps using the same set of presence and background points to convert them into binary maps. This was a necessary calibration step to standardise the scales of predicted suitability. Once the binary maps of at-risk areas were constructed, our map and previously published maps were compared and every $5 \times 5$ km pixel was categorized into four distinct groups: those at-risk in both our map and previous maps, those at-risk only in previous maps, those at-risk only in our map, and areas deemed not at-risk in either.

## Independent validation

As part of the validation process, the initial versions of the suitability maps were presented in June 2023[84] to the Technical Advisory Group on Arbovirus (TAG-Arbovirus[85]), a multidisciplinary group of experts supported by WHO Arbovirus Secretariat. Following the presentation, we invited feedback by providing the TAG with an interactive map of occurrence records for each disease (https://ahyounglim.shinyapps. io/multi_arbo_mapping/) and a concise questionnaire (Supplementary Table 5). This questionnaire addressed specific questions, particularly regarding any discrepancies between our model estimates and their context-specific knowledge, as well as additional drivers of arbovirus risk that should be taken into consideration. This resulted in two main changes to our models: i) identified additional data sources for chikungunya occurrences in Brazil[38]; ii) included additional covariates, such as *Haemagogus janthinomys*[71] for yellow fever model and under-five mortality, government effectiveness, and physicians density for the surveillance capability model (see section "Covariates").

## Reporting summary

Further information on research design is available in the Nature Portfolio Reporting Summary linked to this article.

## Data availability

Disease occurrence data is available from previous publications on dengue[17], chikungunya[18,38], Zika[19], and yellow fever[20]. Additional data was extracted from publicly available sources, including WHO regional outbreak updates (https://www.who.int/emergencies/disease-outbreak-news); ECDC website (https://www.ecdc.europa.eu/en/); and ProMED mail reports (http://www.promedmail.org) and the HealthMap platform (www.healthmap.org). The maps with administrative boundaries were created using public-domain Natural Earth data, accessed via the rnaturalearth package in R[86].

Climate and environmental covariates are freely available from previous publications (GDP[58], urbanisation[59], temperature suitability[66], treatment-seeking for fever[61], dynamic habitat indices[69], and yellow fever vaccination coverage[72]). Surface travel time covariates are available from the Malaria Atlas Project (https://data.malariaatlas.org/maps). Child mortality, physicians density, and the government effectiveness estimates can be freely downloaded via European Commission Disaster Risk Management Knowledge Centre (https://drmkc.jrc.ec.europa.eu/inform-index/). High resolution population data can be freely obtained from LandScan programme (https://landscan.ornl.gov/about). Global climate data can be downloaded from TerraClimate (https://www.climatologylab.org/terraclimate.html). Normalised Difference Vegetation Index data is freely available from NASA Earth Observation Data (https://www.earthdata.nasa.gov/) and can be downloaded using R MODIStp package (https://github.com/ropensci/MODISstp). Predicted suitability for *Ae. albopictus*, Ae. *aegypti, Haemagogus janthinomys*, and non-human primates are not publicly available but can be obtained by contacting the authors of the cited papers[20,69,71]. A detailed description of data sources can be found in Supplementary Tables 1 and 4.

Processed versions of these datasets used in our analyses are available in two repositories: the study github repository (https://github.com/ahyoung-lim/Arbo_riskmaps_public) for past and current versions, and the Figshare repository (https://doi.org/10.6084/m9.figshare.26172934) for the version that has been peer-reviewed and described in this article.

## Code availability

Data analyses were carried out using R version 4.3.0 using the following packages: doParallel (v1.0.17), foreach (v1.5.2), pdp (v0.8.1), matrixStats (v1.3.0), Metrics (v0.1.4), cutpointr (v1.1.2), boot (v1.3-28.1), blockCV (v3.1-3), rsample (v1.2.0), randomForest (v4.7-1.1), terra (v1.7-29), rnaturalearth (v0.3.4), raster (v3.6-23), sp (v1.6-0), exactextractr (v0.9.1), sf

(v1.0-14), tidyterra (v0.5.2), countrycode (v1.5.0), dplyr (v1.1.2), data.table (v1.14.8). All code and processed datasets used for the analyses are publicly available online in the study github repository (https://github.com/ahyoung-lim/Arbo_riskmaps_public) for past and current versions and the Figshare repository (https://doi.org/10.6084/m9.figshare.26172934) for the version that has been peer-reviewed and described in this article.

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

## Acknowledgements

This work was discussed with the Technical Advisory Group on arboviruses (TAG-Arbovirus) and WHO yellow fever risk assessment working group. This work was supported by the World Health Organisation Global Arbovirus Initiative, partially funded through a cooperative agreement with the U.S. CDC. OJB was supported by a UK Medical Research Council Career Development Award (MR/V031112/1) which also supports AL. AL was additionally supported by the Basic Science Research Programme through the National Research Foundation of Korea (NRF) funded by the Ministry of Education (2022R1A6A3A03061207). M.U.G.K. acknowledges funding from The Rockefeller Foundation (PC-2022-POP-005), Google.org, the Oxford Martin School Programmes in Pandemic Genomics & Digital Pandemic Preparedness, European Union's Horizon Europe programme projects MOOD (#874850) and E4Warning (#101086640), Wellcome Trust grants 303666/Z/23/Z, 226052/Z/22/Z & 228186/Z/23/Z, the United Kingdom Research and Innovation (#APP8583), the Medical Research Foundation (MRF-RG-ICCH-2022-100069), UK International Development (301542-403), the Bill & Melinda Gates Foundation (INV-063472) and Novo Nordisk Foundation (NNF24OC0094346). WMdS was supported by a Global Virus Network fellowship, the Burroughs Wellcome Fund (#1022448), and a Wellcome Trust–Digital Technology Development award (Climate Sensitive Infectious Disease Modelling; 226075/Z/22/Z). CJ was supported by a Bloomsbury Colleges PhD Studentship and a National University of Ireland Denis Phelan Scholarship. SJR was supported by NSF CIBR: VectorByte: A Global Informatics Platform for studying the Ecology of Vector-Borne Diseases (NSF DBI 2016265). SJR was additionally supported by funding to Verena (viralemergence.org), including NSF BII 2021909 and NSF BII 2213854. MC was supported by Bill and Melinda Gates Foundation. KAMG received funding from Gavi [Grant ID: 226727_Z_22_Z], BMGF [Grant Numbers INV-034281 and INV-009125 / OPP1157270] and/or the Wellcome Trust via the Vaccine Impact Modelling Consortium during the course of the study. KAMG also acknowledges funding from the MRC Centre for Global Infectious Disease Analysis (reference MR/X020258/1), funded by the UK Medical Research Council (MRC). This UK funded award is carried out in the frame of the Global Health EDCTP3 Joint Undertaking. KAMG reports speaker fees from Sanofi Pasteur outside the submitted work.

## Author contributions

O.J.B. and N.G. conceived and planned the study. A.L., J.C., A.G., C.J., H.K., W.Mde.S., K.S., and J.S.B. identified and obtained data for analysis. A.L., J.C., A.G., C.J., H.K., K.S., and J.S.B. extracted, processed and geopositioned the data. A.L. and O.J.B. carried out the statistical analyses with assistance and input from F.M.S., D.M.P., S.J.R. and N.G. A.L. and O.J.B. prepared the first draft of the manuscript with assistance from F.M.S., K.S., D.M.P., C.J., H.K., J.P.M., M.U.G.K., K.A.M.G., W.Mde.S., E.O.N., M.C., N.F., S.J.R., I.B.R., D.P.R., S.I.H. and N.G.

## Competing interests

The authors declare no competing interests.

## Additional information

[1]Department of Infectious Disease Epidemiology and Dynamics, Faculty of Epidemiology and Population Health, London School of Hygiene & Tropical Medicine, London, UK. [2]Centre for Mathematical Modelling of Infectious Diseases, Faculty of Epidemiology and Population Health, London School of Hygiene & Tropical Medicine, London, UK. [3]Infectious Disease Dynamics Unit, Centre for Epidemiology and Biostatistics, Melbourne School of Population and Global Health, The University of Melbourne, Melbourne, Australia. [4]The Kids Research Institute Australia, Perth Children's Hospital, Perth, Australia. [5]Boston Children's Hospital, Boston, MA, USA. [6]Institute for Health Metrics and Evaluation, University of Washington, Seattle, WA, USA. [7]Department of Health Metrics Sciences, School of Medicine, University of Washington, Seattle, WA, USA. [8]University of Cambridge, Cambridge, UK. [9]Epidemiology Unit, Ministry of Health, Colombo, Sri Lanka. [10]School of Tropical Medicine and Global Health, Nagasaki University, Nagasaki, Japan. [11]School of Geography and the Environment, University of Oxford, Oxford, UK. [12]Oxford School of Global and Area Studies, University of Oxford, Oxford, UK. [13]Department of Biology, University of Oxford, Oxford, UK. [14]Pandemic Sciences Institute, University of Oxford, Oxford, UK. [15]MRC Centre for Global Infectious Disease Analysis, School of Public Health, Imperial College London, London, UK. [16]Department of Microbiology, Immunology and Molecular Genetics, University of Kentucky, College of Medicine, Lexington, KY, USA. [17]Department of Global Health, School of Public Health, Boston University, Boston, MA, USA. [18]Virus Genomic Epidemiology, Faculty of Medicine, School of Public Health, Imperial College, London, Imperial College London, London, UK. [19]Department of Geography and the Emerging Pathogens Institute, University of Florida, Gainesville, FL, USA. [20]Department of Epidemic and Pandemic Preparedness and Prevention, World Health Organization, Geneva, Switzerland. [21]Department of Pediatrics, Harvard Medical School, Boston, MA, USA. [22]Curtin University, Perth, Australia.
✉e-mail: Oliver.Brady@lshtm.ac.uk

