## [Transparent Peer Review file · Nature Communications]

The overlapping global distribution of dengue, chikungunya, Zika and yellow fever

Corresponding Author: Dr Oliver Brady

Version 0:

Reviewer comments:

Reviewer #4

(Remarks to the Author)

I have read the paper for the first time, and I am impressed with the approach, the methods, and how the results are presented.

However, there are some limitations that should be added to the text to acknowledge what the analysis may be lacking, especially in light of the previous suggestions and comments provided by the reviewers.

MAJOR COMMENTS

Overall, I suggest mentioning and discussing four key points in the Discussion as limitations of the proposed approach, or at least highlighting areas where the reader should interpret the results with caution:

1. The geographic overlap of dengue, chikungunya, and Zika is closely linked to the presence of both *Ae albopictus* and *aegypti* in most affected countries. However, this may not apply to countries where only *albopictus* is present, as there are differences in vector competence across these two species in transmitting the considered diseases. This may be particularly relevant for Zika in Europe. Related to this, in European countries where *Ae aegypti* is absent, the likelihood of observing dengue cases is likely lower compared to sympatric areas. [limitation]
2. In non-endemic regions, the risk of disease occurrence is also influenced by the likelihood of disease importation, which is driven by human mobility (more likely in wealthier and more urbanized locations—correlating with GDP and urbanization, as included in the model). However, also the incidence of the disease among travelers is a key factor. Specifically, since dengue has a higher incidence, and yellow fever is limited to Africa and South America, I anticipate differences in the number of travel-related cases and in the risk of local transmission across the diseases. [interpretation]
3. Differences in symptoms and detection across diseases: The differing symptoms and severity associated with various diseases may also play a role. Since the authors assume equivalent detection probabilities between different arboviruses and other viral infections, this limitation should also be acknowledged in the main text. [limitation]
4. The authors state that they assume the long-term environmental niche is static and aim to estimate a long-term average risk. Based on the analysis of a stable environmental niche for Zika in recent decades, the authors suggest that the observed geographic expansion reflects the disease spreading into areas that were always environmentally suitable. These are strong statements that could potentially be misleading. Firstly, how should public health decision-makers interpret these long-term risk maps when defining current surveillance or control efforts? Secondly, given that rapid changes in both the epidemiology of arboviral diseases and environmental conditions have been observed over the last decades, the authors should mention in the limitations that their study does not consider any changing trends (e.g. climate + surveillance + mobility + urbanization + vector presence/competence/resistance, etc.). In particular, it should also be noted that the likelihood of disease occurrence may increase if both vectors are present (e.g., the potential expansion of *Ae aegypti* in Europe). [limitation]

TO BE CLARIFIED IN THE TEXT

The definition of suitability remains somewhat unclear. Please clarify the timeframe for the following statement: "Our measure of suitability is the probability of one or more cases of any of these diseases occurring given a multi-year (2010-2020) average of a location's environmental conditions." Are you referring to 'one or more cases occurring each year' or 'one or more cases over 10 years', ..? Please be specific on this point.

MINOR POINTS

FIGURES:

I suggest highlighting in the text that Figure 3 should be interpreted as representing risk under the assumption of equal surveillance capacity.

RESULTS

The latter part of the claim regarding the northern Sahara is not well supported by Supplementary Figure 7: "with highly populated areas showing higher performance than sparsely populated regions, such as the far north of the Sahara (Supplementary Fig. 7)." Recommendation: Remove "such as the far north of the Sahara (Supplementary Fig. 7)."

Upon reviewing Table 6 in the Supplement, I observe that model performances are quite similar. Therefore, I suggest toning down the statement: "The joint model outperformed individual disease models by offering marginal improvements in performance metrics, particularly benefiting Zika, which has sparse data (Supplementary Table 6)."

Recommendation: I suggest saying that the joint model shows similar performances.

In the section "Comparison with Previous Maps", please specify which model (joint vs. disease-specific) is used for comparing results with other maps (Figure 4).

METHODS:

The sentence "HealthMap is a tool for tracking infectious disease events 15 functional languages by aggregating data from diverse informal online sources" is missing the word "in" after "tracking infectious disease events."

The sentence "...as the RF model has been shown to outperform other methods for presence-only species distribution modelling across a range of applications" repeats a concept mentioned earlier and should be removed.

Something is missing in the sentence after Equation 1

Please include a reference to a dictionary (in the supplement) also for the covariates used in Equation 2.

(Remarks on code availability)

Reviewer #5

(Remarks to the Author)

This manuscript presents a comprehensive global remapping of the likely distributions of several important Aedes-borne diseases.

A major challenge with work like this is to present sufficient details about the data, methods/approaches, and results in a way that is digestible to the readership, while following the necessary word limits, etc. In my opinion that authors have done a commendable job at this - even though it is at times a bit irksome to need to constantly go to supplemental materials to understand what all is going on.

Line 94: "Global maps of disease risk are essential for identifying vulnerable populations..." Maps are clearly important for infectious disease epidemiology, but it is probably a stretch to argue that they're 'essential for identifying vulnerable populations. They could certainly help with identification of populations at risk. Perhaps either give an example or two when identification of vulnerable populations relied on global maps of disease, or consider reworking this sentence.

Line 96: "The spatial distribution of vector-borne disease is defined by the spatial distribution of humans" of course, this is only true with regard to diseases of humans. And even then saying that it is 'defined by' is very absolute. Human population distribution is obviously one major factor with regard to the spatial distribution of vector-borne diseases of humans. Does it completely define it though? Again, please consider softening or offering more support for this very strong statement.

Line 119: "...overcoming the key challenges of observation bias and shared drivers..." I am admittedly not an expert in machine learning, and I do believe the authors have done used some sophisticated approaches to dealing with observation bias in these data (which arises in places where cases might exist but aren't diagnosed or documented). I'm not sure if it is possible to completely overcome these key challenges though.

Related to biases that emerge from a lack of surveillance and even diagnosis, I don't see any discussion about diagnostic approaches and how vastly different diagnostic approaches would figure into this work. Even in many regions that are

dengue endemic, it can be relatively rare to have confirmatory testing done on samples from individuals that have been diagnosed with a given disease (e.g. dengue fever). The ability to even do a confirmatory diagnosis tends to be clustered in urban centers, with access to sophisticated laboratories. In many places, if you go to a clinic or hospital during the rainy season, with symptoms that mostly align with dengue symptoms, then you'll be diagnosed with dengue but never have an actual laboratory test done to confirm it. It is even less likely to do confirmatory testing for chikungunya or Zika. With chikungunya, once it becomes obvious that there is an outbreak of this disease, then the surveillance for that specific disease increases (as healthcare providers become aware that this virus is spreading, and as diagnostics become available). Chikungunya infections in particular behave quite a lot like dengue (and can easily be misdiagnosed as dengue). Zika infections are quite often asymptomatic. Serological tests for this virus will often cross-react with dengue. Some consideration about how these factors must influence our known distribution of these diseases should probably be included.

I do see some mention of these problems in the discussion, but it is not clear to me that these issues have been taken as seriously as they should be. The authors seem to think that "Our approach addresses these by accounting for spatial biases in surveillance and joint modelling of arboviral diseases..." While I would agree that they've attempted to address these issues, I'm not sure it is possible to fully address the issues. There is also mention a few times of how surveillance in the U.S.A. and Europe is more robust. Don't count on that. The ability to do surveillance in wealthy countries like the U.S.A. and some countries of Europe is undoubtedly higher, but that doesn't at all mean surveillance for these types of diseases is actually better. I propose that a physician in Brazil or Thailand, where there is a very long history of experiencing a disease like dengue fever, is much more likely to quickly and accurately diagnose this type of disease than in most parts of the U.S.A or Europe.

Table 1: please provide more description in the table or in its caption. I'm able to make sense of the table when reading the text (which refers to the estimated 5.66 billion people living in areas at risk), but looking at the table alone I don't know what the 5.66 is referring to. I'm also not sure if it makes sense to list 'countries' where >10% of the population are at risk. Perhaps mention this in the table caption too though so that readers can easily figure out what is going on in the table and make their own decisions.

Finally: the spatial resolution of the resulting maps is a limitation. For folks working on the ground, and especially for Aedes borne-diseases which tend to be highly focal (these mosquitoes don't fly much), 5km is a very large space.

(Remarks on code availability)

REVIEWER COMMENTS

Reviewer #4 (Remarks to the Author):

I have read the paper for the first time, and I am impressed with the approach, the methods, and how the results are presented.

However, there are some limitations that should be added to the text to acknowledge what the analysis may be lacking, especially in light of the previous suggestions and comments provided by the reviewers.

MAJOR COMMENTS

Overall, I suggest mentioning and discussing four key points in the Discussion as limitations of the proposed approach, or at least highlighting areas where the reader should interpret the results with caution:

1. The geographic overlap of dengue, chikungunya, and Zika is closely linked to the presence of both *Ae albopictus* and *aegypti* in most affected countries. However, this may not apply to countries where only *albopictus* is present, as there are differences in vector competence across these two species in transmitting the considered diseases. This may be particularly relevant for Zika in Europe. Related to this, in European countries where *Ae aegypti* is absent, the likelihood of observing dengue cases is likely lower compared to sympatric areas. [limitation]

We agree with the reviewer that *Aedes aegypti* and *Aedes albopictus* have different vector competencies for dengue, chikungunya and Zika viruses that may be consequential for transmission efficiency and eventual disease burden. However, both vectors are capable of transmission in the laboratory and areas where only *Ae. albopictus* is present have observed transmission of all three diseases¹ suggesting such areas are at risk (what our maps and models aim to estimate) even if resulting incidence varies (not what our map estimates and potentially not linearly correlated). Our modelling approach includes specific distribution maps for *Ae. aegypti* and *Ae. albopictus* and by allowing for interactions between disease and presence or absence of each of these mosquito species should, in theory, be able to capture any virus-vector species differences that are consequential for the global distributions of these diseases. We do not, however, see strong evidence for this in our specific analysis. We have added an additional sentence to the methods to clarify that our approach is capable of accounting for different virus-vector species relationships:

“This allows the model to share information between arboviral diseases, but also generate distinct relationships between risk of each disease and the environmental covariates if they are justified², e.g. different vector competencies for different viruses”

2. In non-endemic regions, the risk of disease occurrence is also influenced by the likelihood of disease importation, which is driven by human mobility (more likely in wealthier and more urbanized locations—correlating with GDP and urbanization, as included in the model). However, also the incidence of the disease among travelers is a key factor. Specifically, since dengue has a higher incidence, and yellow fever is limited to Africa and South America, I anticipate differences in the number of travel-related cases and in the risk of local transmission across the diseases. [interpretation]

We agree that this is an important limitation that we have now elaborated on in the discussion:

“A key limitation of these maps is that they show all areas suitable for transmission not where these viruses are currently circulating which is likely to be more constrained by viral dispersal. Pairing these maps with estimates of human mobility from endemic areas could allow more specific estimates of areas currently experiencing transmission.”

3. Differences in symptoms and detection across diseases: The differing symptoms and severity associated with various diseases may also play a role. Since the authors assume equivalent detection probabilities between different arboviruses and other viral infections, this limitation should also be acknowledged in the main text. [limitation]

Yes a good point. We now state in the Discussion:

“Our surveillance model may also be influenced by the spatial distribution of each of the viral diseases used in the training data, limiting our ability to isolate pure surveillance effort. This model also assumed the equivalent detection probabilities across different viral infections despite their different clinical presentations and methods of diagnosis. To minimise these effects we chose a broad range of viral diseases with different global distributions, defining surveillance capability as the wider investment in detecting, diagnosing, and reporting any circulating pathogens, but future studies could incorporate more direct measures of surveillance, like surveillance system assessments³ or comparisons between reported case data and serologically-detected infections⁴.”

4. The authors state that they assume the long-term environmental niche is static and aim to estimate a long-term average risk. Based on the analysis of a stable environmental niche for Zika in recent decades, the authors suggest that the observed geographic expansion reflects the disease spreading into areas that were always environmentally suitable. These are strong statements that could potentially be misleading. Firstly, how should public health decision-makers interpret these long-term risk maps when defining current surveillance or control efforts? Secondly, given that rapid changes in both the epidemiology of arboviral diseases and environmental conditions have been observed over the last decades, the authors should mention in the limitations that their study does not consider any changing trends (e.g. climate + surveillance + mobility + urbanization + vector presence/competence/resistance, etc.). In particular, it should also be noted that the likelihood of disease occurrence may increase if both vectors are present (e.g., the potential expansion of *Ae aegypti* in Europe). [limitation]

Yes these are all good points.

We now include an additional limitation on shifting risk over time in the limitations paragraph of the discussion:

“Past and future changes in climate, urbanisation, surveillance capacity and spread of arboviral vectors could all change the resulting risk maps. Due to data sparsity at the global level it is necessary to aggregate data over multiple years, limiting our ability to capture these changes in global risk over time. While previous work has found that such changes will only have minimal and gradual impacts on the environmental niche for dengue^{5,6}, tipping point events, such as invasion of *Ae. aegypti* into mainland Europe could lead to more rapid changes in risk necessitating updating of these maps.”

We have also included a new paragraph at the end of the discussion directed towards users of these maps and explicitly mention time frames and updates over time as important needs:

“Our suitability maps offer a starting point for national to local-scale risk assessments over the short to medium term, helping to target interventions and surveillance efforts. These maps should be used in combination with local risk mapping studies that utilise fine-scale spatial information on arboviral disease cases and mosquito counts to further target locally. New data generated from such efforts could be used to iteratively update these maps to improve their contemporariness and operational relevance, contributing to more effective disease control efforts worldwide.”

TO BE CLARIFIED IN THE TEXT

The definition of suitability remains somewhat unclear. Please clarify the timeframe for the following statement: “Our measure of suitability is the probability of one or more cases of any of these diseases occurring given a multi-year (2010-2020) average of a location’s environmental conditions.” Are you referring to ‘one or more cases occurring each year’ or ‘one or more cases over 10 years’, ..? Please be specific on this point.

To clarify, the occurrence data used in this study indicate whether a disease has ever been reported at a location up to March 2024, while the environmental predictors were averaged over the 10-year period (2010–2020) which corresponds to the timeframe when most occurrence points are concentrated.

We have revised this as follows: “Our measure of suitability is probability of one or more cases of any of these diseases ever having occurred up to 2024, based on the average environmental conditions of a location over a multi-year period (2010-2020).”

MINOR POINTS

FIGURES:

I suggest highlighting in the text that Figure 3 should be interpreted as representing risk under the assumption of equal surveillance capacity.

We thank reviewer for the valuable suggestion.

We now describe this approach in the figure caption:

“Fig. 1. Model-predicted environmental suitability for dengue, chikungunya, and Zika (a), and yellow fever (b) after accounting for spatial variation in surveillance capacity. Areas without a suitable temperature range for transmission have been set to 0 for dengue, chikungunya, and Zika. Areas outside the countries at risk, endemic, or potentially at risk for yellow fever as defined by the WHO yellow fever risk assessment working group⁷ have been set to 0.”

RESULTS

The latter part of the claim regarding the northern Sahara is not well supported by Supplementary Figure 7: “with highly populated areas showing higher performance than sparsely populated regions, such as the far north of the Sahara (Supplementary Fig. 7).”

Recommendation: Remove “such as the far north of the Sahara (Supplementary Fig. 7).”

We thank reviewer for bringing this to our attention. The latter part has now been removed.

Upon reviewing Table 6 in the Supplement, I observe that model performances are quite similar. Therefore, I suggest toning down the statement: “The joint model outperformed individual disease models by offering marginal improvements in performance metrics, particularly benefiting Zika, which has sparse data (Supplementary Table 6).”

Recommendation: I suggest saying that the joint model shows similar performances.

We have changed this to: “The joint model showed comparable performance to individual disease models with slight improvements in some metrics, particularly benefiting Zika, which has sparse data (Supplementary Table 6).”

In the section “Comparison with Previous Maps”, please specify which model (joint vs. disease-specific) is used for comparing results with other maps (Figure 4).

To clarify, the joint modelling approach was used throughout the manuscript, while disease-specific models were tested only for performance comparisons against the joint models. We have now made this distinction more explicit in the “Comparison with previous maps” subsection:

“Our joint models generally predict a more focal distribution of dengue, chikungunya, and Zika than previous maps, ...”

and also in the “Sensitivity analyses” subsection:

“To assess the benefits of joint modelling over disease-specific models, we re-ran individual models for dengue, chikungunya, Zika, and yellow fever separately and compared the model performance metrics (AUC, sensitivity, and specificity) with those of the joint model. Model performance was evaluated using 10-fold cross-validation with metrics calculated based on a cutoff value where the sum of sensitivity and specificity is maximised.”

METHODS:

The sentence “HealthMap is a tool for tracking infectious disease events 15 functional languages by aggregating data from diverse informal online sources” is missing the word “in” after “tracking infectious disease events.”

Done

The sentence “...as the RF model has been shown to outperform other methods for presence-only species distribution modelling across a range of applications” repeats a concept mentioned earlier and should be removed.

Done

Something is missing in the sentence after Equation 1

Apologies for this oversight. We have now corrected this after Equation 1:

“ $P(\text{occ}_{viral_i})$ is the location-specific probability that a record i is an occurrence record of an emerging acute febrile disease, rather than a background point, and f denotes the non-linear and interacting function of all covariates at the same spatial location as record i .”

Please include a reference to a dictionary (in the supplement) also for the covariates used in Equation 2.

We have included the dictionary in the Supplementary table 4.

###

Reviewer #5 (Remarks to the Author):

This manuscript presents a comprehensive global remapping of the likely distributions of several important Aedes-borne diseases.

A major challenge with work like this is to present sufficient details about the data, methods/approaches, and results in a way that is digestible to the readership, while following the necessary word limits, etc. In my opinion that authors have done a commendable job at this - even though it is at times a bit irksome to need to constantly go to supplemental materials to understand what all is going on.

We thank reviewer for the positive comment.

Line 94: "Global maps of disease risk are essential for identifying vulnerable populations..." Maps are clearly important for infectious disease epidemiology, but it is probably a stretch to argue that they're 'essential for identifying vulnerable populations. They could certainly help with identification of populations at risk. Perhaps either give an example or two when identification of vulnerable populations relied on global maps of disease, or consider reworking this sentence.

We appreciate the reviewer's valuable feedback. We agree that global maps of disease risk are more appropriately framed as tools for identifying populations at risk, rather than being "essential" for identifying vulnerable populations. We have revised the sentence to better reflect this:

"Global maps of disease risk are valuable tools for identifying vulnerable populations, guiding surveillance, and maximising the impact and efficiency of surveillance and control efforts."

Line 96: "The spatial distribution of vector-borne disease is defined by the spatial distribution of humans" of course, this is only true with regard to diseases of humans. And even then saying that it is 'defined by' is very absolute. Human population distribution is obviously one major factor with regard to the spatial distribution of vector-borne diseases of humans. Does it completely define it though? Again, please consider softening or offering more support for this very strong statement.

We agree that the statement may be too absolute and have now soften the sentence:

"The spatial distribution of a vector-borne disease in humans is influenced by the spatial distribution of humans, vectors, environmental features that shape transmission efficiency and mobility that spreads the pathogen to suitable areas."

Line 119: "...overcoming the key challenges of observation bias and shared drivers..." I am admittedly not an expert in machine learning, and I do believe the authors have done used some sophisticated approaches to dealing with observation bias in these data (which arises in places where cases might exist but aren't diagnosed or documented). I'm not sure if it is possible to completely overcome these key challenges though.

We agree with the reviewer that it is not possible to completely overcome these challenges. We now state :

“Our approach builds on an existing ecological niche modelling (ENM) framework, and attempts to address the key challenges of observation bias and shared drivers by...”

Related to biases that emerge from a lack of surveillance and even diagnosis, I don't see any discussion about diagnostic approaches and how vastly different diagnostic approaches would figure into this work. Even in many regions that are dengue endemic, it can be relatively rare to have confirmatory testing done on samples from individuals that have been diagnosed with a given disease (e.g. dengue fever). The ability to even do a confirmatory diagnosis tends to be clustered in urban centers, with access to sophisticated laboratories. In many places, if you go to a clinic or hospital during the rainy season, with symptoms that mostly align with dengue symptoms, then you'll be diagnosed with dengue but never have an actual laboratory test done to confirm it. It is even less likely to do confirmatory testing for chikungunya or Zika. With chikungunya, once it becomes obvious that there is an outbreak of this disease, then the surveillance for that specific disease increases (as healthcare providers become aware that this virus is spreading, and as diagnostics become available). Chikungunya infections in particular behave quite a lot like dengue (and can easily be misdiagnosed as dengue). Zika infections are quite often asymptomatic. Serological tests for this virus will often cross-react with dengue. Some consideration about how these factors must influence our known distribution of these diseases should probably be included.

I do see some mention of these problems in the discussion, but it is not clear to me that these issues have been taken as seriously as they should be. The authors seem to think that "Our approach addresses these by accounting for spatial biases in surveillance and joint modelling of arboviral diseases..." While I would agree that they've attempted to address these issues, I'm not sure it is possible to fully address the issues. There is also mention a few times of how surveillance in the U.S.A. and Europe is more robust. Don't count on that. The ability to do surveillance in wealthy countries like the U.S.A. and some countries of Europe is undoubtedly higher, but that doesn't at all mean surveillance for these types of diseases is actually better. I propose that a physician in Brazil or Thailand, where there is a very long history of experiencing a disease like dengue fever, is much more likely to quickly and accurately diagnose this type of disease than in most parts of the U.S.A or Europe.

We appreciate the several pertinent points the reviewer makes about the different dimensions of surveillance biases.

With regards to spatial variation in access to laboratory testing facilities, we aim to account for this using the surveillance model fit to occurrence point data from all viral diseases. By pooling data across a wide range of viral conditions over long time periods nearly every inhabited area should have experienced some notifiable acute viral diseases. Therefore the remaining spatial pattern effectively estimates the relative probability that any viral disease will be diagnosed and reported. Our estimate of “surveillance capacity” includes factors such treatment seeking, propensity to report and access to laboratory testing facilities. Unfortunately, disaggregating surveillance capacity down into its constituent parts was not possible in this analysis.

With regards to misdiagnosis among arboviruses, we agree that this is likely common and one of the reasons we prefer to use occurrence over incidence data due to the diminished

influence of misdiagnosis. To clarify, our approach to using occurrence data only assigns a single occurrence point to each 5x5km pixel on the map-ie has Zika every been reported here. While we agree, as you say, that symptom-based diagnosis of cases is likely common once the outbreak has begun, this likely follows diagnosis of a case with more reliable laboratory methods. The situation where a single case is laboratory diagnosed and all subsequent cases are symptom-based and a situation where all cases are laboratory diagnosed would each generate just a single occurrence point in our analysis. If data were available at this scale on method of confirmation we could perform a sensitivity analysis on the modes fit to only laboratory-confirmed cases, however this was not possible and, based on the previous statement we would expect minimal impact.

With regards to the impact of physician familiarity we agree that this is likely to have a big impact in the early stages of dengue emergence in new settings. The first cases of dengue in areas with general strong surveillance systems are likely to be undiagnosed, however, over the longer term once this risk has been appropriately identified and diagnostic practices appropriate adjusted we are confident that surveillance for dengue will reach a comparable standard to other acute viral diseases. We have included an additional sentence in the “approach” section of the methods to clarify this:

“The surveillance model aims to capture the between and within country differences in the long-term average probability that a person infected with an acute viral infection seeks treatment, is correctly diagnosed and is reported in the public domain. It should be noted that such estimates may not correlate with ability to detect unfamiliar newly emerging diseases but will represent longer-term differences in surveillance capacity once locally circulating diseases have been appropriately characterised.”

Table 1: please provide more description in the table or in its caption. I'm able to make sense of the table when reading the text (which refers to the estimated 5.66 billion people living in areas at risk), but looking at the table alone I don't know what the 5.66 is referring to. I'm also not sure if it makes sense to list 'countries' where >10% of the population are at risk. Perhaps mention this in the table caption too though so that readers can easily figure out what is going on in the table and make their own decisions.

Thank you for the suggestion We have made slight revisions to Table 1 to improve readability and have added more detailed captions.

Table 1. Estimated global population and number of countries at risk for dengue, chikungunya, Zika, and yellow fever.

Region	Dengue, chikungunya and Zika	Yellow fever
Population at risk (in billions)*		
Global	5.66 (5.64-5.68)	1.54 (1.53-1.54)
Africa	1.24 (1.24-1.24)	1.14 (1.14-1.15)
Asia	3.46 (3.45-3.47)	-
Americas	0.73 (0.73-0.73)	0.38 (0.38-0.38)
Europe	0.17 (0.17-0.17)	-
Oceania	0.03 (0.03-0.03)	-

Number of countries (sovereign states) at risk**		
Global	169	54

* Values show mean population estimates with 95% confidence intervals in parentheses. Surveillance capability scores and at-risk population estimates with 95% confidence intervals for each country are available in the Supplementary Data.

** Countries at risk are defined as UN sovereign states where more than 10% of the population is estimated to be living in at-risk areas.

Finally: the spatial resolution of the resulting maps is a limitation. For folks working on the ground, and especially for *Aedes* borne-diseases which tend to be highly focal (these mosquitoes don't fly much), 5km is a very large space.

While we acknowledge that the 5 km resolution may be too coarse for more localised assessments especially for *Aedes*-borne infections, these maps do have strategic value at the national scale for resource prioritisation. We view these maps as starting point for risk assessments at more local scales. We would argue that these maps provide useful information for identifying regions that may be at risk and can serve as a tool to guide further monitoring and investigation. To refine local risk stratifications, they can be combined with more detailed, localised data and resources, such as entomological surveys.

We now state in the last paragraph of the Discussion section that:

“Our suitability maps offer a starting point for national to local-scale risk assessments over the short to medium term, helping to target interventions and surveillance efforts. These maps should be used in combination with local risk mapping studies that utilise fine-scale spatial information on arboviral disease cases and mosquito counts to further target locally. New data generated from such efforts could be used to iteratively update these maps to improve their contemporariness and operational relevance, contributing to more effective disease control efforts worldwide.”

References

1. Brady, O. J. & Hay, S. I. The first local cases of Zika virus in Europe. *Lancet* **394**, 1991–1992 (2019).
2. Shearer, F. M. *et al.* Estimating Geographical Variation in the Risk of Zoonotic *Plasmodium knowlesi* Infection in Countries Eliminating Malaria. *PLoS Negl. Trop. Dis.* **10**, e0004915 (2016).
3. World Health Organization. *Surveillance and control of arboviral diseases in the WHO African region: assessment of country capacities.* <https://www.who.int/publications/i/item/9789240052918> (2022).

4. Stanaway, J. D. *et al.* The global burden of dengue: an analysis from the Global Burden of Disease Study 2013. *Lancet Infect. Dis.* **16**, 712–723 (2016).
5. Harish, V. *et al.* Human movement and environmental barriers shape the emergence of dengue. *Nat. Commun.* **15**, 4205 (2024).
6. Messina, J. P. *et al.* The current and future global distribution and population at risk of dengue. *Nat. Microbiol.* **4**, 1508–1515 (2019).
7. Eliminate Yellow fever Epidemics (EYE): a global strategy, 2017–2026. *Wkly Epidemiol Rec* **92**, 193–204 (2017).